# Correlation between seismic activity and acoustic emission on the basis of in-situ monitoring

**Zhiwen Zhu**[1]; **Zihan Jiang**[1]; **Federico Accornero**[1]; **and Alberto Carpinteri**[1]

[1]Department of Civil Engineering and Intelligent Construction, Shantou University, 243 University Road, 515063 Shantou, China

**Correspondence:** Zhiwen Zhu (zhuzw@stu.edu.cn)

**Abstract:** Since April 2023 an in-situ experimental campaign has started at a granite underground tunnel, which is a dedicated monitoring platform located in Southeast China. Acoustic Emission (AE) signals and seismic sequences were simultaneously recorded by installing the AE device together with the seismometer, in order to investigate, among other parameters, the $b$-value and the natural-time variance, $\kappa_1$, of AE time series. In addition, AE and related temporal correlation with the incoming seismic events are analyzed using an appropriate multi-modal statistical analysis. The results show that AE has a strong correlation with seismic swarms in surrounding areas. The changing trend of AE temporal distribution occurs before that of the earthquake and regularly anticipates the seismic major event by approximately 17 hours. The AE bursts indicate that an earthquake is approaching. The dense clusters of AE are closely related to two major earthquakes with Richter magnitudes equal to 3.2 and 2.4. Approaching the earthquake occurrence, the $b$-value shows a downward trend, reaching its minimum value prior to the earthquake, whereas the natural-time variance $\kappa_1$ rapidly decreases from 0.07 to a minimum value close to zero. $\kappa_1$ occurs earlier than the minimum $b$-value and the AE bursts. Therefore, trends of the $b$-value and the natural-time variance derived from the AE time series can be used as effective earthquake precursors. It is also evident that there is widespread micro-seismic activity in the earthquake preparation zone before the earthquake occurrence. The micro-seismic activity represents the origin of microcracks in the nearby ground surface, resulting in the AE bursts. The results of this paper provide new experimental evidence for the application of fracto-emissions as seismic precursors.

## 1. Introduction

Earthquake precursors are phenomena that take place in advance before the occurrence of an earthquake. These precursors are various, such as ground deformation and stress, changes in Earth tidal strain, geoacoustic and geomagnetic fields, environmental radioactivity, and so on. In the time period before the earthquake occurrence, a very wide area of cracking rocks is active around the epicentre of the upcoming earthquake (Carpinteri et al., 2017). Solids that break in a brittle way are subjected to a rapid emission of mechanical energy, involving the generation of pressure waves that travel at a characteristic speed with the order of magnitude of $10^3$ m/s. Assuming a constant pressure wave velocity, the correlation between the wavelength (forming crack) scale and the frequency scale is shown in Fig.1. The frequency range of pressure waves is very wide, from nano-scale defects emitting at the frequency scale of Terahertz ($10^{12}$ Hz), to kilometer-scale fractures emitting at the scale of Hertz (Carpinteri et al., 2017), which is a typical frequency of seismic oscillations and can be detected by sensors arranged on solid bodies. In the framework of Fracture Mechanics, Acoustic Emission (AE) represents a specific part of the strain energy released during the damage process, and it is emitted in the form of transient ultrasonic waves. Prior to an earthquake, AE bursts may result from widespread micro-seismic activity, which propagates through the ground.

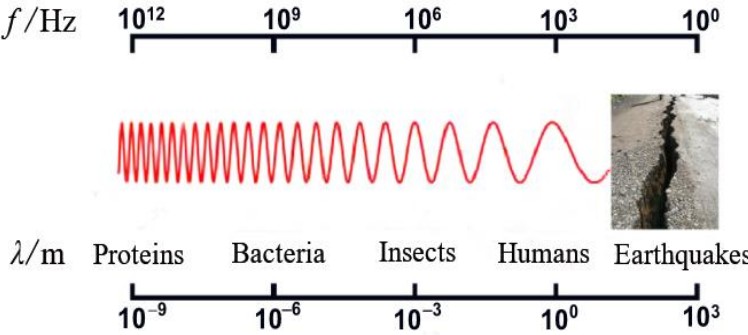

Figure 1: Correlation between wavelength scale and frequency scale (Modified from Carpinteri 2017).

Moreover, according to recent interpretations (Lacidogna et al., 2019), the relationship between crack propagation and emitted energy is represented by the areas subtended by the snap-back instability branches in the load vs displacement diagram (Fig.2). In Fig.2, the grey areas identify the dissipated energy, D, whereas the pink ones represent the emitted energy, E. The total energy released during the loading process, R, is equal to the sum of the dissipated energy, D, plus emitted energy, E. When an earthquake occurs, AE does not come only from a single macro-crack, but from a wide network of microcracks generated by micro-seismic activity before the earthquake occurrence. In this way, AE can be effectively used as a seismic precursor.

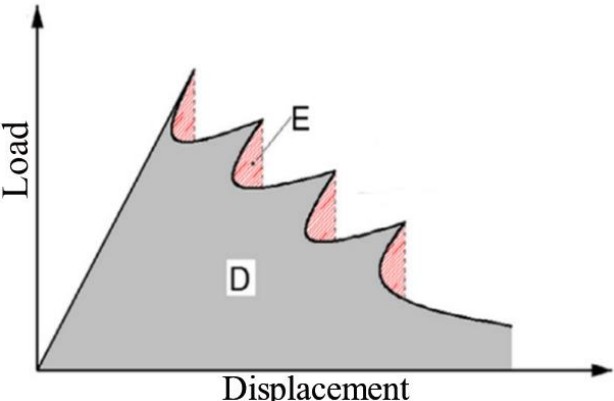

Figure 2: Multiple local instabilities (snap-back) caused by micro-seismic activity (Modified from Carpinteri 2016a).

An important aspect of earthquake prediction is the dimension and temporal evolution of the "earthquake preparation zone". Dobrovolsky et al. (1979) assumed that the "strain zone" is a circle with the centre located at the epicentre of the impending earthquake. The radius $R$ of the circle is called strain radius and is assumed to be a function of the magnitude of the upcoming earthquake. As an example, for an earthquake with a Richter magnitude M equal to 6, the strain radius is about 1000 kilometers. Carpinteri et al. (2019) proposed an innovative estimation of the earthquake preparation zone, which is proportional to the magnitude of the upcoming earthquake and depends on the average size of the cracks forming in the Earth's crust before the seismic event. Approaching the earthquake occurrence, this zone tends to shrink and the pre-existing and external cracks close, forming a new and smaller preparation zone where the remaining open cracks coalesce to form larger cracks. Therefore, when the earthquake eventually occurs, the preparation zone will coincide with the earthquake epicentre (Carpinteri et al., 2019).

As shown in Fig.3, in the earlier stages of a seismic event, the preparation zone develops from its maximum size, and, during the first stage, nano- and micro-cracks dominate. In the following stage, as the tectonic stresses tend to get closer to the earthquake epicentre, the preparation zone will shrink, and the average crack size will increase from the micro-scale to the

millimetre-scale. Approaching the earthquake occurrence, a further size reduction of the preparation zone is expected, characterized by larger cracks, from the millimetre-scale to the metre-scale, which are able to generate ultrasonic acoustic waves up to several hundreds of kHz. At the final stage, the macro-cracks along the seismic fault will coalesce and the earthquake will take place accompanied by audible acoustic emission. Figure 3 shows the evolution of the preparation zone proposed by Carpinteri et al. (2019): each circle represents the evolution of the strain zone. The equivalent crack sizes in the subsequent areas are: $10^{-9} \sim 10^{-6}$ m (purple), $10^{-6} \sim 10^{-3}$ m (blue), $10^{-3} \sim 1$ m (red), and the black dot is the epicentre of the final earthquake.

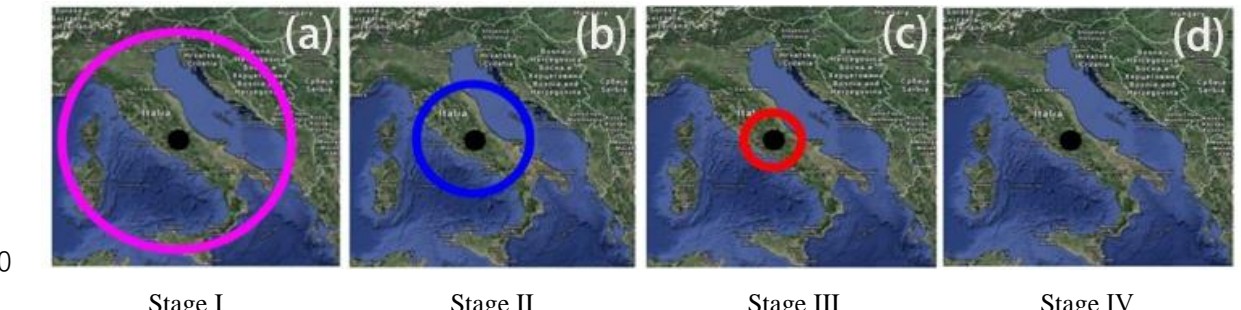

|  Stage I | Stage II | Stage III | Stage IV |

Figure 3: Evolution of earthquake preparation zone (Modified from Carpinteri 2019).

## 2. AE as seismic precursor

Nowadays, the AE technique is widely used in the field of structural monitoring for civil engineering (Manuello et al., 2019; Han et al., 2019; Dong et al., 2019). In addition, AE can be used as a diagnostic tool in geophysics (Olga et al., 2022; Moriya et al., 2018; Yuri et al., 2023). There are crustal stresses and strains widely distributed within the preparation zone of seismic events (Gregori et al., 2010), and the AE bursts may be interpreted as a characteristic of the crustal stress redistribution (Lacidogna et al., 2011; Carpinteri et al., 2016b). AE can be regarded as an earthquake precursor. For example, before the Assisi earthquake, a large and almost sudden burst in AE was observed about 400 kilometers away from the epicentre (Gregori et al., 2005), confirming the fact that AE can be used for earthquake prediction. The correlation between the AE activity in masonry structures and regional earthquakes has been studied by the AE technique (Carpinteri et al., 2013). One of the authors (Lacidogna et al., 2015) proposed a new procedure for earthquake risk assessment based on AE technology, developing statistical methods for space-time correlations between AE and seismic events. In order to evaluate the propagation of in-situ stresses, Zimator et al. (2017) used AE time series obtained from two monitoring stations located about 300 kilometers far apart in Italy. It was found that AE can identify anomalies on crustal stress trend, which may be related to earthquake occurrence. Carpinteri et al. (2017; 2019) installed a monitoring station in a gypsum mine located in northern Italy, where experimental observations show that AE is strongly related to earthquakes occurring in the surrounding area.

In addition, $b$-value and natural-time variance of AE time series can be used as earthquake precursors. The importance of the $b$-value for quantifying seismic activity (Allen et al., 1965) or earthquake prediction (Smith et al., 1981) has been widely recognized by seismologists. Sammonds et al. (1992) found that the $b$-value shows a V-shaped curve with a significant decrement before a major earthquake, whereas most seismic activities occur near the minimum value of the $b$-value curve. Han et al. (2015) proposed a robust method for estimating the $b$-value and found that, compared to two traditional methods, this method provides reliable $b$-values and shows good sensitivity to the large-magnitude earthquakes. In addition, the natural-time analysis was proposed only few years ago, and few applications to earthquake prediction are currently presented in the scientific literature. Varotsos et al. (2001) found that, before the occurrence of an earthquake, the seismic activity located in the same tectonic zone would enter a critical stage, whereas the natural-time variance $\kappa_1$ fluctuates around the critical value

0.07, decreasing quickly to zero when the earthquake occurs. Sarlis et al. (2013) conducted a study in Japan (25°N~46°N, 125°E~148°E), considering all the earthquakes with magnitude M ≥ 3.5 during the time interval 1984-2011. The results show that for the earthquakes with M ≥ 7.6, $\kappa_1$ shows a minimum value before the earthquake occurrence.

Since April 2023, an in-situ experimental campaign has started in a granite underground tunnel located in Southeast China. AE signals and seismic sequences were simultaneously recorded, and the AE temporal correlation with the incoming seismic events is analyzed using multi-modal statistical analysis. In addition, the *b*-value and the natural-time variance of AE time series were further investigated. Through the use of the seismometer installed together with the AE device, seismic signals from nearby areas are monitored to record micro-seismic events. Since the monitoring centre is located in a dedicated tunnel, the impact of the environmental ultrasonic noise has been eliminated, such as that coming from traffic, human activities, and wind.

## 3. In-situ monitoring of AE and earthquakes

An AE and earthquake monitoring system is arranged inside a dedicated tunnel of the seismic monitoring centre of Shantou City. This centre is the backbone station for comprehensive seismic observation in the Eastern region of Guangdong Province, in the Southeast of China. It is located at latitude 23.415°, North, and longitude 116.628°, East (Fig.4). This all-granite tunnel is excavated horizontally into the mountain up to 150 m, and is mainly used to install seismic observation instruments, detecting seismic data such as crustal deformation and underground fluids. All the seismic station equipment is connected to the National Seismic Monitoring Network. This dedicated tunnel is far away from the noise sources of traffic, human activities, and wind noise, minimising the interference of external environmental factors. In addition, the soil and tunnel structure provide sound-absorbing and sound-insulating effects, reducing external noise transmission.

The AE equipment employed is the ÆMISSION system, as shown in Fig.5. The eight-channel system stores signal parameters, including duration, rise time, energy, amplitude, and ringing count, allowing a continuous AE monitoring for the desired time period. This monitoring device uses eight AE sensors (frequency range 10 kHz - 1 MHz) fixed on the ground surface, together with a seismometer to monitor the seismic activity. In order to ensure a great accuracy in data collection, eight AE sensors are placed in the same location. This allows for the comparison and verification of the consistency of the monitoring results, thus ensuring the absence of errors or missed detections.As mentioned above, AE sources (microcracks coming from micro-seismic activity) are widespread in a very large area prior to the earthquake. Thus, AE event source localisation is not crucial in the framework of the multi-modal statistical analysis. On the contrary, with the current sensor setup, which has no source localisation capability due to the sensor positions, our recordings reflect the AE activity of all channels as a whole. In the post-processing stage, AE signals with a duration shorter than 3 μs and containing less than three oscillations across the detection threshold were discarded, filtering out electrical noise. The monitoring started at 12:21 on April 24, 2023, and ended at 11:10 on May 29, 2023, resulting in a continuous monitoring for 35 days (839 hours).

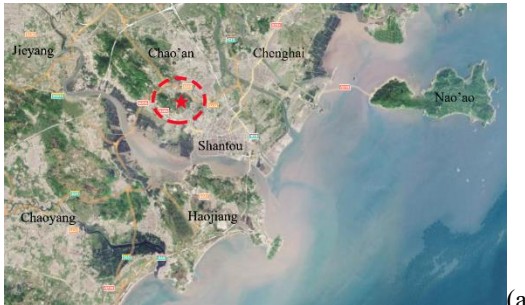
(a)
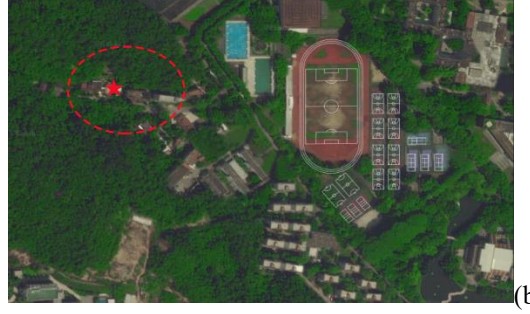
(b)

Figure 4: Location of seismic monitoring centre. (a) Shantou seismic station; (b) Detail location of dedicated tunnel.

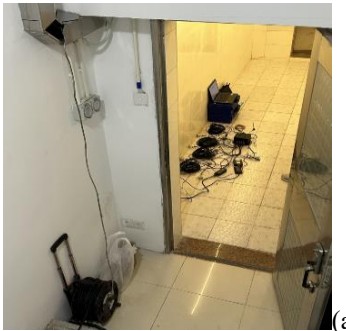 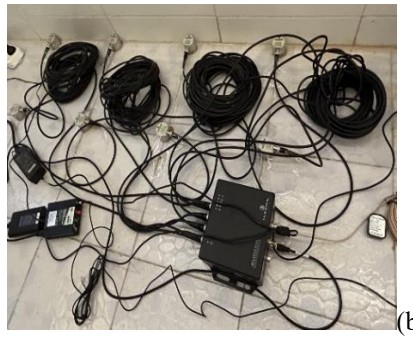 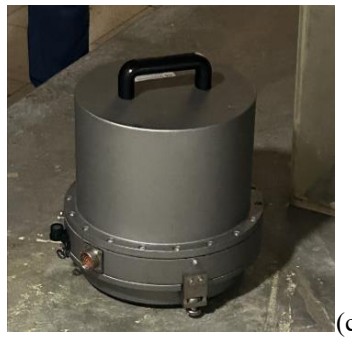

(a) (b) (c)

Figure 5: Set-up of monitoring system.(a) Interior view of tunnel; (b) AE acquisition device; (c) Seismometer.

## 4. Correlation between seismic and AE activities

### 4.1 Regional seismic activity

Shantou City is a strong earthquake zone in China. The Taiwan Strait, located in the Southeast of Shantou City, is a strong earthquake-prone area with frequently observed moderate to strong earthquakes. The largest reported earthquake in Shantou
history occurred in 1918, with a Richter magnitude of 7.9 and epicentre in Nan'ao, an island close to the coast in the South China Sea. The historical seismic activity in the surrounding areas of Shantou is shown in Fig.6. A total of 51 earthquakes with magnitude M ≥ 4.7 were recorded in the region starting from the year 1067 to 2022, including two earthquakes with magnitudes 7.0-7.9, ten earthquakes with magnitudes 6.0-6.9, 22 earthquakes with magnitudes 5.0-5.9, and 17 earthquakes with magnitudes 4.7-4.9. Since the establishment of the Guangdong Provincial Seismic Network in 1970, only one earthquake
of magnitude M ≥ 4.0 has been recorded in the near-field region, i.e., within 25 kilometers from Shantou, which is the earthquake of magnitude 4.2 in the Chenghai District, occurred on January 16, 2004. Historically, there are three other far-field strong earthquakes near the area: the Quanzhou Earthquake of magnitude 7.1 in 1604, the Haifeng Earthquake of magnitude 6.0 in 1911, and the Taiwan Strait Earthquake of 7.3 magnitude in 1994. In Fig.6, the seismic intensity in this region is generally high. It is worth noting that the vast majority of destructive earthquakes are distributed in the coastal areas of the
Eastern continent.

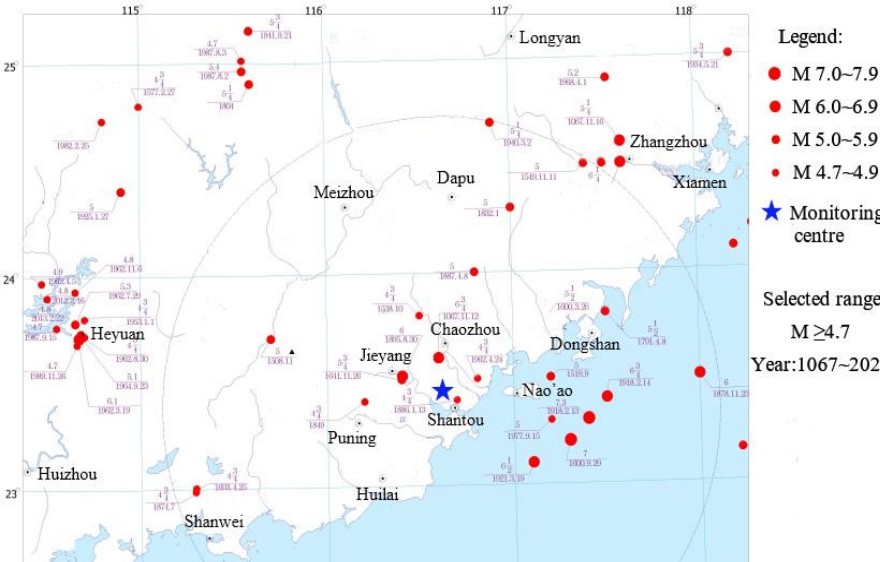

Figure 6: Historical seismic activity in area surrounding Shantou City, China.

In consideration of the historical seismic activity in the surrounding area, as well as the scale (Dobrovolsky et al., 1979) and

time evolution (Carpinteri et al., 2019) of the "earthquake preparation zone" in the pre-earthquake time period, regional

earthquakes with epicentre within 200 kilometers from the monitoring centre were selected. During the monitoring time period,

small seismic events took place in the area frequently. Based on the magnitude and epicentre distance, two major regional

earthquakes are selected: Richter magnitude 3.2 earthquake in Taiwan Strait at 23:50 Beijing Time on April 30, 2023 (EQ.1

for short), and Richter magnitude 2.4 earthquake in Haifeng Sea Area of Guangdong Province at 12:23 Beijing Time on May

17, 2023 (EQ.2 for short). The satellite map showing the location of the two major earthquake is reported in Fig.7. EQ.1 (red

dot) has the epicentre 160.1 kilometers far from the monitoring centre (blue dot), whereas EQ.2 (red dot) has the epicentre

132.4 kilometers far from the monitoring centre (blue dot). The earthquake information is shown in Table 1.

Table 1. Information about two major earthquakes

| Richter Magnitude ($M_L$) | Date of occurrence | Beijing time | Epicentral distance (km) | Epicentre location | North latitude (°) | East longitude (°) |
|---|---|---|---|---|---|---|
| 3.2 | April 30, 2023 | 23:50 | 160.1 | Taiwan Strait | 23.37 | 118.57 |
| 2.4 | May 17, 2023 | 12:23 | 132.4 | Haifeng Sea Area | 22.83 | 115.24 |

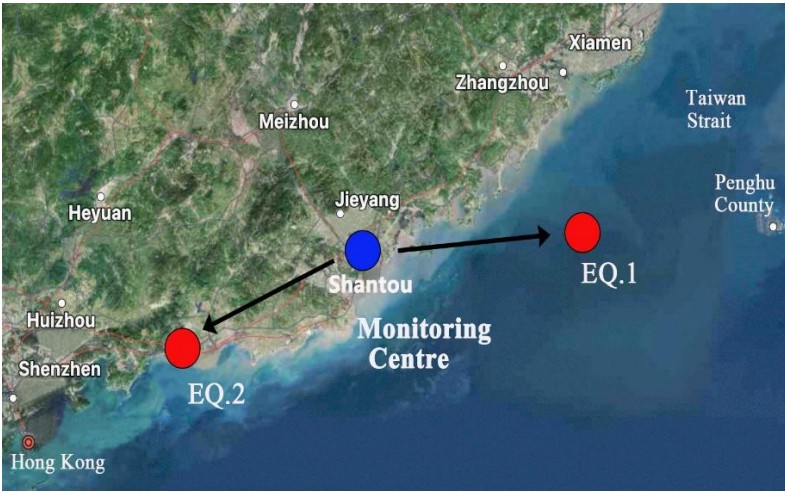

Figure 7: Satellite map of epicentres of two major earthquakes.

## 4.2 AE time series and seismic activity

The seismic sequence and the AE time series, including the cumulated AE and the AE rate, are shown in Fig.8. The AE rate represents the number of AE events per hour. The blue dots in Fig.8 are the Richter magnitudes ($M_L$) of the seismic events detected during the monitoring period. The time correlation between AE clusters and seismic events can be observed throughout the monitoring time period. The dense clusters of AE signals, especially around 142 h and 566 h, show significant peaks in the AE rate, which appear to anticipate the earthquakes with Richter magnitude equal to 3.2 (EQ.1) and Richter magnitude 2.4 (EQ.2). The cumulated AE represents the total number of AE events occurred during the monitoring period. The times marked by red stars in Fig.8 are $t_{EQ.1}$ (April 30, 2023) and $t_{EQ.2}$ (May 17, 2023), which represent the occurrence times of the two major earthquakes. The seismic events are anticipated by large jumps in the cumulated AE and by significant peaks in the AE rate.

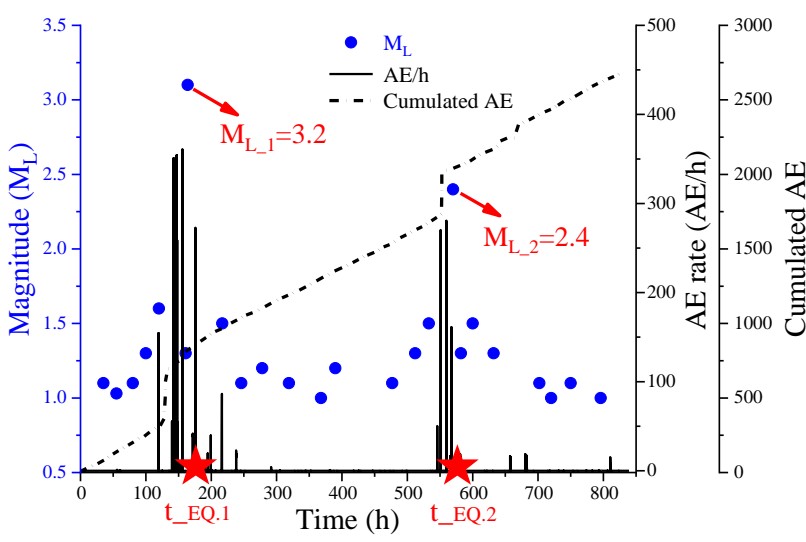

Figure 8: AE rate and cumulated distribution versus seismic sequence.

## 4.3 Multi-modal statistical analysis

A multi-modal (Gaussian and multi-peak) statistical analysis is carried out by means of Microcal Origin, identifying the relative maxima of AE and seismic distributions by best Gaussian fitting. The optimal Gaussian fitting reproduces all the peaks, minimising the gap between the predicted values and the actual data.    In particular, starting from the discrete distribution of data and following an iterative procedure, in which the curve offset, $y_0$, centre coordinate, $x_c$, width, $w$, and amplitude, $A$, are considered, the multi-modal curve that best approximates the discrete distribution of points is identified by the following

equations (Fig.9):

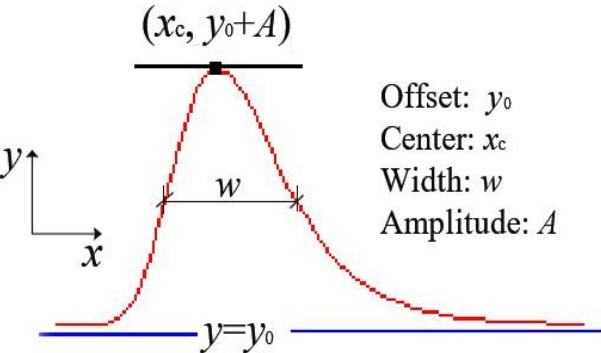

Figure 9: Parameters of the Gaussian curve.

$$y = y_0 + Ae^{(-e^{-z}-z+1)}, \tag{1}$$

$$z = \frac{x - x_c}{w}. \tag{2}$$

    This multi-modal approach is used for the statistical analysis of earthquakes and AE temporal distributions. Regarding the temporal distribution of the 24 earthquakes detected during the 35-day monitoring time period, two major seismic swarms are identified in Fig.10(a,b), together with the two AE Gaussian distributions.

The superposition of AE and earthquake distributions is shown in Fig.10(c), where it is evident the strong correlation between seismic swarms occurring in the monitored area and AE signals, as well as the precursor role played by AE with respect to imminent earthquakes. The Gaussian fitting parameters employed to plot Fig.10(c) are the following: Centre coordinate ($x_c$): 148, 165, 556, 573; Curve offset ($y_0$): 22, 1.1, 22, 1.1; Amplitude ($A$): 339.0, 2.0, 261.3, 1.3; Width ($w$): 32.8, 24.0, 32.8, 19.0.

Figure 10(d,e) clearly shows how AE anticipates each seismic event by approximately 17 hours for both the events.

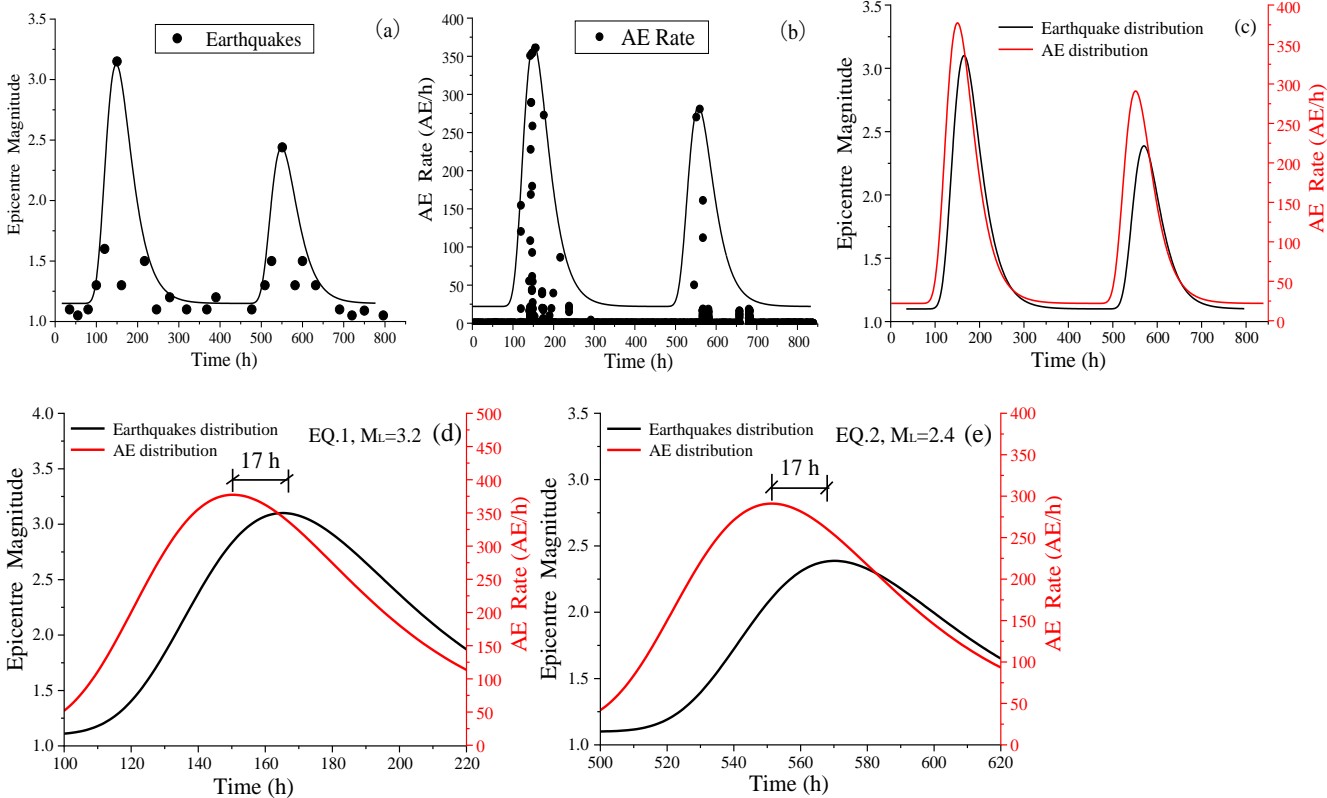

Figure 10: Multi-modal Gaussian distribution of earthquakes (a), AE (b), superposition (c) and the predicted results (d),(e).

### 4.4 b-value and seismic sequences

Although seismic events exhibit complex space-time behaviour, reflecting the extreme disorder of Earth's crust, universally valid scaling laws emerge. The *b*-value is able to describe the evolution of seismic events by considering the statistical distribution of AE signal magnitude following the Gutenberg–Richter (GR) law (Carpinteri et al., 2011). The GR law was

215 firstly introduced in seismology and then extended to the statistics of AE signals (Colombo et al., 2003):

$$\log_{10}(N_{AE}) = a - bM \,, \tag{3}$$

Where $M = \log_{10}(A_{max})$, $A_{max}$ is the signal peak amplitude, and $N_{AE}$ is the number of AE events with magnitude greater than $M$. The *b*-value is the negative slope of the GR law straight line, which is fitted by the least squares method. In this study, the temporal variation of the *b*-value is estimated by the moving event window method. A number of events, $N$, equal to 400,

and a time window step of 200 events are adopted for the evaluation of the *b*-value temporal variation.

In Fig.11, the AE *b*-values and seismic magnitudes of events EQ.1 and EQ.2 are reported. The occurrence of two major earthquakes resulted in a significant decrement in the *b*-value, reaching a minimum below 1. When approaching the $M_{L\_1}$=3.2 earthquake, the *b*-value drops from 2.3 to 0.9. Then, approaching the $M_{L\_2}$=2.4 earthquake, the *b*-value drops from 2.1 to 0.9. On the other hand, when no major earthquakes occur, the *b*-value tends to increase again. The downward trend of *b*-value can

be used as an early warning of earthquakes, the time of minimum *b*-value being just prior to the earthquake occurrence.

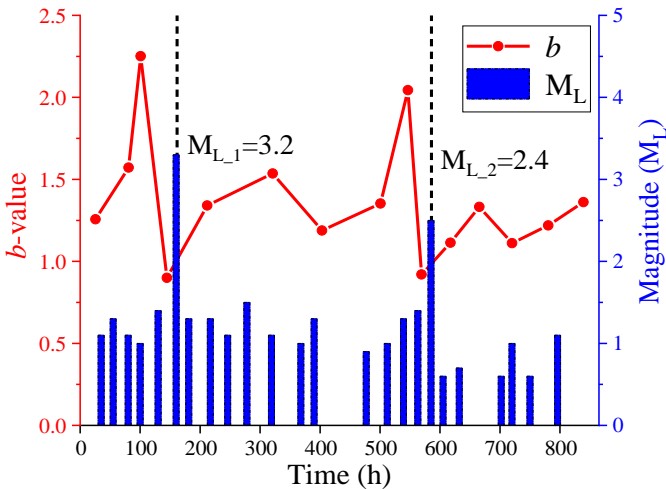

Figure 11: The *b*-value of AE and magnitudes of earthquakes.

### 4.5 Natural-time analysis

Recently, natural time analysis has been applied to identify the imminent failure of materials and structures (Loukidis et al., 2020; Ferreira et al., 2022a and 2022b; Triantis et al., 2023). Natural-time series transform time series into the natural-time domain, ignoring the time intervals of consecutive events and only considering the order and energy of occurrence. Based on the time-series analysis of $N$ events read in a new time domain, namely the natural time, $\chi$ , a method to identify critical states was developed (Varotsos et al., 2011 and 2013). The variance $\kappa_1$ of the natural time is defined as:

$$\kappa_1 = \sum_{k=1}^{N} p_k \chi_k^{\,2} - \left( \sum_{k=1}^{N} p_k \chi_k \right)^2 = \left\langle \chi^2 \right\rangle - \left\langle \chi \right\rangle^2 , \tag{4}$$

Where $\chi_K = K / N$ is a normalized index of of energy $Q_k$ (related AE energy), and $P_K = Q_K / \sum_{i=1}^{N} Q_i$ is the probability distribution of the discrete variable $\chi_K$ . When $\kappa_1$ converges to 0.07, the critical state is imminent.

In particular, two conditions have been defined to identify the entrance of the monitored structure to a true critical state (Vallianatos et al., 2013):

(I) The parameter $\kappa_1$ approaching the value 0.07 "by descending from above".

(II) The entropies $S$ and $S_{rev}$ lower than the entropy of the uniform noise, which is $S_u$=0.0966. The entropy s is defined as:

$$S = \left\langle \chi \ln \chi \right\rangle - \left\langle \chi \right\rangle \ln \left\langle \chi \right\rangle, \quad \text{where} \left\langle \chi \ln \chi \right\rangle = \sum_{k=1}^{N} \chi_k \ln \chi_k . \tag{5}$$

Similarly, the entropy $S_{rev}$ is obtained by considering the time reversal $T_{PK} = P_{N\_K+1}$ .

Therefore, when critical conditions (I) and (II) are satisfied, the moment at which critical state occurs can be justified (Hloupis et al., 2015 and 2016).

Hence, the evolution of variance $\kappa_1$, entropy $S$, and $S_{rev}$ of natural-time series $\{\chi_k\}$ for the seismic event EQ.1 is reported in

Fig.12, showing the three variables ($\kappa_1$, $S$, $S_{rev}$) as functions of time t. The two horizontal dotted lines represent the variance limit $\kappa_1$=0.07 and the entropy limit $S_u$=0.0966, respectively. According to the critical conditions (I) and (II), the value of $t_{crit}$ is shown in Fig.15. In particular, $t_{\_EQ.1}$ represents the occurrence time of the earthquake EQ.1. When approaching the earthquake occurrence, the natural-time variance $\kappa_1$ rapidly decreases from 0.07 to a minimum value close to zero.

In addition, a comparison between $t_{crit}$ and $t_{b\text{-}min}$, which is the time characterising the minimum $b$-value, shows a substantial agreement between the two indicators (see Table 2). The critical time of the variance being earlier than the time of minimum $b$-value and the time of AE cluster. Thus, the trends of $b$-value and natural-time variance can be used as seismic precursors.

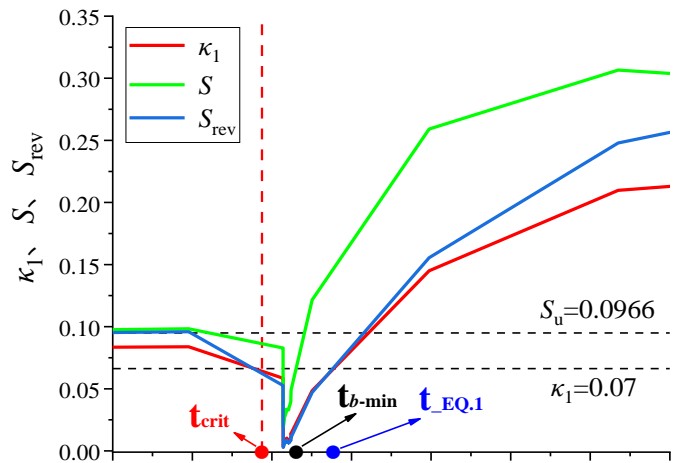

Figure 12: Natural-time series and seismic events.

Table 2. AE anticipating the seismic event EQ.1

| Critical time $t_{crit}$ [h] | Time of AE cluster [h] | Time of minimum $b$-value, $t_{b\text{-}min}$ [h] | Time of the earthquake occurrence $t_{\_EQ.1}$ [h] |
|---|---|---|---|
| 139 | 142 | 146 | 155 |

### 4.6 AE parameters approaching the major earthquake

In order to analyze AE parameters before the major earthquake (EQ.1), the AE time series between 135 and 155 hours are taken, focusing on the representative waveforms and b-values during this time period. The AE parameters coming from the acoustic waves, as shown in Fig.13, where the ringing count is the number of signal oscillations greater than the AE threshold, the duration is the time elapsed between the first and the last signal oscillation above the AE threshold, the amplitude is the signal peak amplitude, and the average frequency is calculated as the ringing counts divided by the duration.

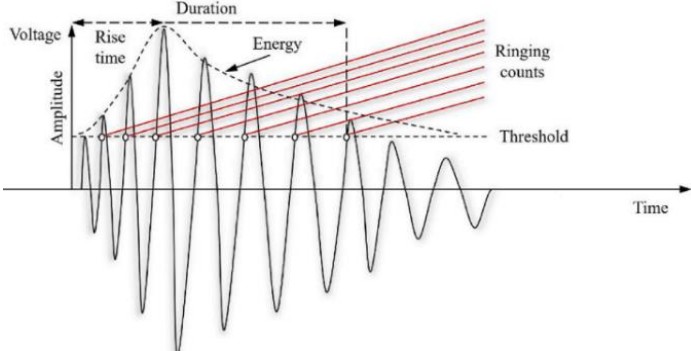

Figure 13: Acoustic wave parameters.

The densest AE cluster approaching the earthquake occurrence is shown in Fig.14, presenting the following characteristics: a large jump in the cumulated AE and significant peaks in AE rate, frequency and amplitude. It is worth noting that this AE cluster is closely related to the Richter magnitude 3.2 earthquake (EQ.1), appearing about 13 hours in advance. When approaching the earthquake occurrence, there is an AE burst. This can be explained by the fact that there was extensive micro-seismic activity in the earthquake preparation zone before the earthquake occurrence, which may have caused the generation of micro-cracks in the nearby ground surface.

In addition, as seen from the *b*-value analysis in Fig.14, the temporal variation of the *b*-value is estimated by the moving event window. A number of events, N, equal to 20, and a time window step of 10 events are adopted for the evaluation of the *b*-value temporal variation. The *b*-value continuously decreases when approaching the earthquake and then reaches the minimum *b*-value, which is 9 hours earlier than the occurrence time of the earthquake. Moreover, the continuous downward trend of the *b*-value can be used as an early warning of earthquakes.

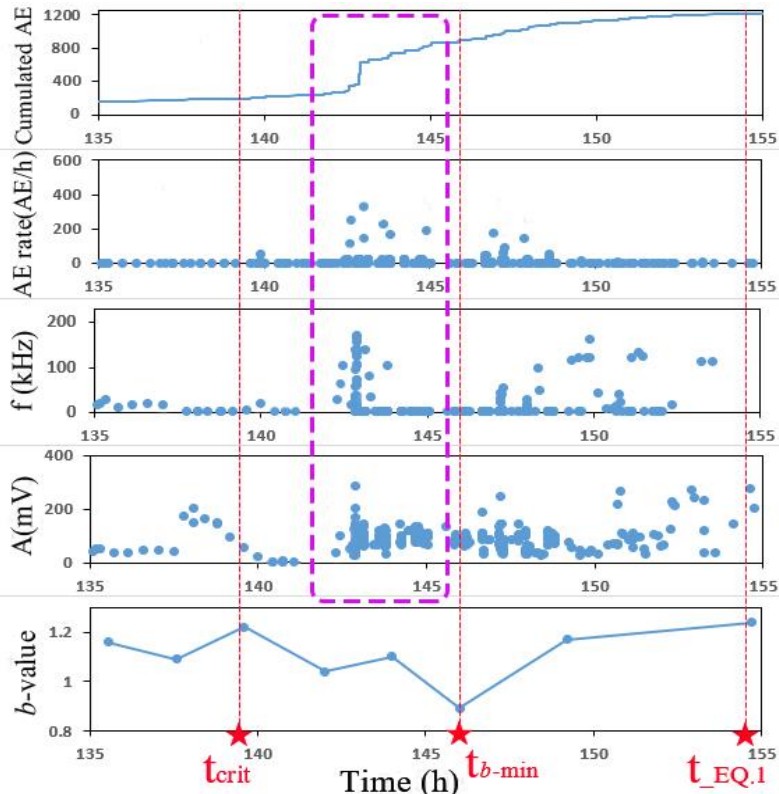

Fig.14: AE parameters and *b*-value when approaching the major earthquake occurrence. $t_{crit}$ represents the critical time as predicted by the natural-time analysis, $t_{\_EQ.1}$ depicts the occurrence time of earthquake EQ.1.

## 5.Conclusions and future perspectives

Since April 2023, an in-situ experimental campaign has started at a granite underground tunnel located in Southeast China, revealing the strong seismic forecasting potentialities of the AE peaks by means of a dedicated monitoring platform. The AE and its temporal correlation to the incoming seismic events are analyzed using an appropriate multi-modal statistical analysis. The conclusions are as follows:

(1) The monitoring equipment is arranged in the granite underground tunnel with low noise, and the ideal AE monitoring data and ground motion data are obtained, which indicates the reasonable feasibility of the monitoring system in this paper.

(2) The dense clusters of AE, especially around 142 h and 566 h, show large jumps in the cumulated AE and significant peaks in the AE rate, which appear to anticipate the earthquakes with Richter magnitude 3.2 (EQ.1) and the Richter magnitude 2.4 (EQ.2). There was extensive micro-seismic activity before the earthquake occurrence, which may represent the origin of microcracks in the nearby ground surface, resulting in the AE bursts. Thus, AE can be used as seismic precursor.

(3) Multi-modal statistical analysis of earthquake and AE distributions shows that AE has a strong correlation to seismic swarms occurring in surrounding areas. The evaluation trend of AE temporal distribution develops prior to that of the earthquake, and AE tends to anticipate the next seismic peak with an evident and chronologically ordered shifting, which regularly anticipates by approximately 17 hours both the considered seismic events.

(4) $b$-value analysis shows that, when approaching major seismic events, the $b$-value decreases significantly and reaches a minimum value below 1, revealing that a larger magnitude event is approaching. The downward trend of $b$-value can be used as an early warning of earthquakes, the time of minimum $b$-value being just prior to the earthquake occurrence.

(5) When approaching the earthquake occurrence, the natural-time variance $\kappa_1$ rapidly decreases from 0.07 to a minimum value close to zero, the critical time of the variance being earlier than the time of minimum $b$-value and the time of AE cluster. Thus, the trends of $b$-value and natural-time variance can be used as seismic precursors.

Current research can only provide time predictions of AE as earthquake precursors, not determining the epicentre and the magnitude of the earthquake. Future studies could identify seismic epicentre and magnitude through the networking of multiple AE devices in different locations. In addition, some other precursory phenomena, such as electromagnetic and neutron emissions, could also be analyzed at the dedicated monitoring platform to provide a more accurate basis for earthquake prediction.

**Credit authorship contribution statement**

**Zhiwen Zhu**: Writing – review & editing, Methodology, Investigation, Data curation, Conceptualization. **Zihan Jiang**: Writing – review & editing, Writing – original draft, Methodology, Investigation, Data curation, Conceptualization. **Federico Accornero**: Methodology, Data curation, Conceptualization, Visualization. **Alberto Carpinteri**: Methodology, Supervision, Writing-Review & Editing.

**Declaration of competing interest**

The authors declare that they have no known competing financial interests or personal relationships that could have appeared to influence the work reported in this paper.

**Data availability statement**

Data supporting the research obtainable from the corresponding author upon reasonable request.

**Acknowledgments**

Zhiwen Zhu acknowledges the support from National Natural Science Foundation of China (52278509) and Natural Science Foundation of Guangdong Province (2022A1515 010261). Federico Accornero acknowledges the support from STU Outstanding Talent Grant N. 140-09423016.

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
