# Peer review of "Correlation between seismic activity and acoustic emission on the basis of in-situ monitoring"

_EGUsphere, 2024_

## Author Comment (AC1)

**REPLY TO REVIEWERS**

**Journal:** NHESS

**Manuscript Number:** EGUSPHERE-2024-688

**Title:** Correlation between seismic activity and acoustic emission on the basis of in-situ monitoring

**Authors:** Zhiwen ZHU, Zihan JIANG, Federico ACCORNERO, and Alberto CARPINTERI

The authors thank the Editor and the Reviewers for their useful remarks. The recommendations helped the authors in preparing an improved version of the manuscript.

Concerning the list of comments, a detailed answer is provided below. All the changes required are highlighted with a yellow background in the revised manuscript.

**REVIEWER 1**

*In this study, an in-situ experimental campaign was conducted on a granite underground tunnel located in Southeast China. The objective was to analyze precursor parameters employed on AE time series and its relationship with seismic events recorded. The AE and its temporal correlation to the incoming seismic events were analyzed by considering the multi-modal statistical analysis, b-value, and the natural-time variance. The research topic is important from the point of view of phase transition phenomena and mining engineering. However, some points should be improved.*

*It is recommended that the present manuscript be accepted following the implementation of significant revisions. Some general observations to improve the manuscript clarity and quality are listed:*

*(1) Figure 2 is similar to the one presented in [x1], therefore, it is recommended to clarify that this figure is modified or redrawn to avoid copyright issues.*

*[x1] https://doi.org/10.1016/j.engfracmech.2016.01.013*

Following the Reviewer's suggestion, the manuscript has been revised as follows:

<< …

[Figure]

Figure 2: Multiple snap-back instabilities (Modified from Carpinteri et al. 2016).

…>>

*(2) Some sentences can be added about the recent application of natural time on AE time series in order to identify the imminent failure of materials and structures. The review and comments should cover more new studies, such as those presented in [x2-x5], but not limited to these.*

*[x2] https://doi.org/10.1016/j.physa.2019.123831*

*[x3] https://doi.org/10.3390/app12083918*

*[x4] https://doi.org/10.3390/app12041980*

*[x5] https://doi.org/10.3390/app13106261*

Following the Reviewer's suggestion, the manuscript has been revised as follows:

<<…Recently, natural time analysis has been applied to identify the imminent failure of materials and structures (Loukidis et al., 2020; Ferreira et al., 2022a and 2022b; Triantis et al., 2023)…>>

<<

[32]Loukidis, A., Pasiou, E. D., Sarlis, N. V., and Triantis, D. Fracture analysis of typical construction materials in natural time, Physica A, 547, 123831. https://doi.org/ 10.1016/j.physa.2019.123831, 2020.

[33]Ferreira, L. F., Rojo, N. B. T., Bordin, A. C., Sobczyk, M., Lacidogna, G., Niccolini, G., and Iturrioz, I. Analysis of acoustic emission activity during progressive failure in heterogeneous materials:

experimental and numerical investigation, Appl Sci-Basel, 12, 3918-3918. https://doi.org/10.1016/j.physa.2019.123831, 2022a.

[34]Ferreira, L. F., Silva, É. C., Bordin, A. C., Rojo, N. B. T., Sobczyk, M., Lacidogna, G., and Ignacio, I. Long-range correlations and natural time series analyses from acoustic emission signals, Appl Sci-Basel, 12, 1980. https://doi.org/10.3390/app12041980, 2022b.

[35]Triantis, D., Stavrakas, I., Loukidis, A., Pasiou, E. D., and Kourkoulis, S. K. A study on the fracture of cementitious materials in terms of the rate of acoustic emissions in the natural time domain, Appl Sci-Basel, 13, https://doi.org/10.3390/app13106261, 2023. >>

*(3) In order to clarify the article, please provide more details on how "the impact of environmental ultrasonic noise, such as that from traffic, human activities and wind" has been eliminated.*

The site of the experiment is an all-granite dedicated underground tunnel, which is far away from the noise sources of traffic, human activities, wind noise, etc. This dedicated tunnel is a seismic observation platform located in southeastern China, deep into the mountain up to 150 m. In addition, the soil and tunnel structure provide sound-absorbing and sound-insulating effects, reducing external noise transmission. By considering the Reviewer's suggestion, the manuscript has been revised as follows:

<<…This all-granite tunnel is excavated horizontally into the mountain up to 150 m, and is mainly used to install seismic observation instruments, detecting seismic data such as crustal deformation and underground fluids. All the seismic station equipment is connected to the National Seismic Monitoring Network. This dedicated tunnel is far away from the noise sources of traffic, human activities, and wind noise, minimising the interference of external environmental factors. In addition, the soil and tunnel structure provide sound-absorbing and sound-insulating effects, reducing external noise transmission. >>

*(4) Please provide the criteria used to identify the optimal Gaussian fit in the multimodal analysis and the parameters of the Gaussian fit used to plot Figure 12. This information is relevant for other researchers wishing to apply a similar approach.*

Following the Reviewer's suggestion, the manuscript has been revised as follows:

<< A multi-modal (Gaussian and multi-peak) statistical analysis is carried out by means of Microcal Origin, identifying the relative maxima of AE and seismic distributions by best Gaussian fitting. The optimal Gaussian fitting reproduces all the peaks, minimising the gap between the predicted values and the actual data. In particular, starting from the discrete distribution of data and following an iterative procedure, in which the curve offset, $y_0$, centre coordinate, $x_c$, width, $w$, and amplitude, $A$, are considered, the multi-modal curve that best approximates the discrete distribution of points is identified by the following equations (Fig.10). >>

<< The superposition of AE and earthquake distributions is shown in Fig.10(c), where it is evident the strong correlation between seismic swarms occurring in the monitored area and AE signals, as well as the precursor role played by AE with respect to imminent earthquakes. The Gaussian fitting parameters employed to plot Fig.10(c) are the following: Centre coordinate ($x_c$): 148, 165, 556, 573; Curve offset ($y_0$): 22, 1.1, 22, 1.1; Amplitude ($A$): 339.0, 2.0, 261.3, 1.3; Width ($w$): 32.8, 24.0, 32.8, 19.0. >>

*(5) Please clarify the methodology employed to calculate the b-value. Was a moving event window employed, or was a time window considered?*

The manuscript has been revised as follows:

<<…The *b*-value is the negative slope of the GR law straight line, which is fitted by the least squares method. In this study, the temporal variation of the *b*-value is estimated by the moving event window method. A number of events, *N*, equal to 400, and a time window step of 200 events are adopted for the evaluation of the *b*-value temporal variation. …>>

*(6) It is known that several parameters can be applied to the energy term (P_K) in the analysis of natural time, such as amplitude, rise angle and AE energy (see, for example, Refs [x2-x5]). Which parameter was used by the authors? Please clarify.*

In our study, we calculated the AE energy for each event and used it as the $P_K$ value for analysing the natural time, thus revealing the trend and the characteristics of the system in critical state. Natural-time series transform time series into the natural-time domain neglecting the time intervals of consecutive events, only considering the order and energy of occurrence. Among $N$ events, the energy of events is defined as $Q_k$, $\chi$ is defined as normalized index of energy $Q_k$, $\chi_K = K/N$ Normalized energy $P_K = Q_K / \sum_{i=1}^{N} Q_i$ is a probability distribution of discrete variable $\chi_K$. The text of the manuscript has been revised as follows:

<<…Based on the time-series analysis of $N$ events read in a new time domain, namely the natural time, $\chi$, a method to identify critical states was developed (Varotsos et al., 2011 and 2013). The variance $\kappa_1$ of the natural time is defined as:

$$\kappa_1 = \sum_{k=1}^{N} p_k \chi_k{}^2 - \left( \sum_{k=1}^{N} p_k \chi_k \right)^2 = \langle \chi^2 \rangle - \langle \chi \rangle^2, \qquad (4)$$

where $\chi_K = K/N$ is the normalized index of energy $Q_k$ (related AE energy), and $P_K = Q_K / \sum_{i=1}^{N} Q_i$ is the probability distribution of the discrete variable $\chi_K$. When $\kappa_1$ converges to 0.07, the critical state is imminent. >>

*(7) Why is there no critical time for EQ.2 in the natural time analysis? Even if the natural time parameters do not converge to EQ.2, a future study can be proposed for this.*

The main focus of the present study is on the multi-modal statistical analysis, which proves to be an effective tool for earthquake prediction. In addition, by means of the natural time analysis, we identify the critical time for EQ.1 event. Although we did not find the critical time of the EQ.2 event with

natural time analysis, future works could improve the natural time algorithms to accommodate a wider range of seismic intensities and to increase the accuracy of the critical time prediction. Meanwhile, we need to further collect and analyse more experimental data from stronger earthquakes to determine the effectiveness of the natural time method, which, in this work, could be considered merely as a confirmation of the multi-modal statistical analysis.

---

## Author Comment (AC2)

**REPLY TO REVIEWERS**

**Journal:** NHESS

**Manuscript Number:** EGUSPHERE-2024-688

**Title:** Correlation between seismic activity and acoustic emission on the basis of in-situ monitoring

**Authors:** Zhiwen ZHU, Zihan JIANG, Federico ACCORNERO, and Alberto CARPINTERI

The authors thank the Editor and the Reviewers for their useful remarks. The recommendations helped the authors in preparing an improved version of the manuscript.

Concerning the list of comments, a detailed answer is provided below. All the changes required are highlighted with a yellow background in the revised manuscript.

**REVIEWER 2**

*In their submitted manuscript, the authors addressed the very interesting issue of earthquake precursors, in the context of simultaneously recording of the Acoustic Emission (AE) activity and seismic activity, over a short period of time (35 days) in a case study area, located in southern China. However, the simultaneous monitoring of AE and seismic activity has been reported in the past by one of the authors, namely A. Carpinteri and his coworkers, in a different studied area (see Ref 1.2 and 4) with a quite similar analysis, based on multi-modal statistical analysis (Gaussian fittings of AE rate). Noteworthy that, additional fracto-emission phenomena, such as EM radiation and neutron emission had been considered in those cases (Ref.1). In that sense, the present work, in my opinion, does not contribute any additional knowledge and innovation to the important subject of earthquake precursors, such as the physical mechanism of AE preceding the seismic activity, and/or an advanced data analysis method. In the latter case, the b-value and the natural time analyses that carried out in AE data are quite common methods in the literature that may "predict" the occurrence of critical states in complex systems, such as earthquake and AE series (see for example related papers from Vallianatos and his coworkers, Triantis and his coworkers, etc).*

It is important to remark that the main focus of this work is not on b-value analysis or natural time analysis, which can be here considered as a mere confirmation of the results obtained by means of the multi-modal statistical analysis. On the contrary, the multi-modal analysis is a novelty in the scientific literature on earthquake prediction.

We believe that our study remains unique in that we have analysed data for a specific region and time period, which is of utmost importance in terms of spatial and temporal variability of earthquake precursors.

*Some important issues that the authors should consider are the following:*

*(1) Figures 1, 2 and 3 have been reported exactly the same in the authors' previous works, i.e., in Refs 1, 2 and 4.*

By considering the Reviewer's suggestion, the figures of the manuscript have been revised as follows:

<<

[Figure]

Figure 1: Correlation between wavelength scale and frequency scale (Modified from Carpinteri 2017).

[Figure]

Figure 2: Multiple local instabilities (snap-back) caused by micro-seismic activity (Modified from Carpinteri 2016).

[Figure]

|  |  |  |  |
|---|---|---|---|
| Stage I | Stage II | Stage III | Stage IV |

Figure 3: Evolution of earthquake preparation zone (Modified from Carpinteri 2019). >>

*(2) The monitoring period is quite short, with only 2 major earthquake events. It should be expanded further to include more EQ events and become feasible b-value and natural time analysis, in both datasets, i.e. AE data and regional seismic catalogue. The latter correlation would be very important.*

We fully recognise the importance of the monitoring period, particularly in terms of collecting a large number of seismic events to carry out associated data analysis. In the future we will carry out long-term monitoring here to ensure that more seismic events are included and more extensive data analysis will be carried out, including correlation analysis between the AE data and the regional seismic catalogue.

*(3) A set of 8 AE sensors are used to monitor AE activity but all of them are located at the same position, next to each other. What is the point of using all these AE sensors? It would be meaningful to locate them at different sites of the so called "preparation zone" and not at the same point, just to*

*increase the recorded AE rate.*

We agree with the Reviewer that a layout that covers a wider preparation zone by placing AE sensors at different locations will better capture the seismic activity. We are currently setting up 5~6 different stations and networking them, also including specific equipment for neutron and electromagnetic emissions, along with long-term monitoring.

Moreover, the choice of using many AE sensors is driven by considerations of accuracy in data collection and obtaining comprehensive data. In daily monitoring, multiple sensors are placed in the same location to compare and verify the consistency of their monitoring results to ensure that there are no errors or missed detections. The text of the manuscript has been revised as follows:

<<…The AE equipment employed is the ÆMISSION system, as shown in Fig.5. The eight-channel system stores signal parameters, including duration, rise time, energy, amplitude, and ringing count, allowing a continuous AE monitoring for the desired time period. This monitoring device uses eight AE sensors (frequency range 10 kHz - 1 MHz) fixed on the ground surface, together with a seismometer to monitor the seismic activity. ==In order to ensure a great accuracy in data collection, eight AE sensors are placed in the same location. This allows for the comparison and verification of the consistency of the monitoring results, thus ensuring the absence of errors or missed detections==… >>

*(4) The authors refer to AE events, but how they can be sure that each recorded hit in the 8 channels corresponds to a different generated micro-crack? With the specific arrangement of the 8 sensors (refer to Fig. 5), I believe that source location is not possible, so the recording AE activity is actually the hit rate of all channels. This misunderstanding need to be clarified.*

It is important to remark that the localisation of AE sources in the procedure adopted for earthquake prediction is of secondary importance. As a matter of fact, AE sources (micro-cracks coming from micro-seismic activity) are widespread in a very large area prior to the earthquake. Thus, AE events that are useful to localise the damage source are not crucial in the framework of the multi-modal statistical analysis. On the contrary, with the current sensor setup, which has no source localisation

capability due to the sensor positions, our recordings reflect the AE activity of all channels as a whole.

<< As mentioned above, AE sources (microcracks coming from micro-seismic activity) are widespread in a very large area prior to the earthquake. Thus, AE event source localisation is not crucial in the framework of the multi-modal statistical analysis. On the contrary, with the current sensor setup, which has no source localisation capability due to the sensor positions, our recordings reflect the AE activity of all channels as a whole. In the post-processing stage, AE signals with a duration shorter than 3 μs and containing less than three oscillations across the detection threshold were discarded, filtering out electrical noise. The monitoring started at 12:21 on April 24, 2023, and ended at 11:10 on May 29, 2023, resulting in a continuous monitoring for 35 days (839 hours) >>

*(5) Regarding AE data, only the hit rate is considered and no other useful information is provided. It would be important to include plots of representative AE waveforms and their frequency content, the time evolution of various hit-based AE parameters such energy, duration, average frequency, etc, and not only the hit rate. Actually, the energy rate could be considered as a more reliable AE parameter that the hit rate.*

By considering the Reviewer's suggestion, the manuscript has been revised as follows:

<< **4.6 AE parameters approaching the major earthquake**

In order to analyze AE parameters before the major earthquake (EQ.1), the AE time series between 135 and 155 hours are taken, focusing on the representative waveforms and *b*-values during this time period. The AE parameters coming from the acoustic waves, as shown in Fig.13, where the ringing count is the number of signal oscillations greater than the AE threshold, the duration is the time elapsed between the first and the last signal oscillation above the AE threshold, the amplitude is the signal peak amplitude, and the average frequency is calculated as the ringing counts divided by the duration.

[Figure]

Figure 13: Acoustic wave parameters.

The densest AE cluster approaching the earthquake occurrence is shown in Fig.14, presenting the following characteristics: a large jump in the cumulated AE and significant peaks in AE rate, frequency and amplitude. It is worth noting that this AE cluster is closely related to the Richter magnitude 3.2 earthquake (EQ.1), appearing about 13 hours in advance. When approaching the earthquake occurrence, there is an AE burst. This can be explained by the fact that there was extensive micro-seismic activity in the earthquake preparation zone before the earthquake occurrence, which may have caused the generation of micro-cracks in the nearby ground surface.

In addition, as seen from the $b$-value analysis in Fig.14, the temporal variation of the $b$-value is estimated by the moving event window. A number of events, $N$, equal to 20, and a time window step of 10 events are adopted for the evaluation of the $b$-value temporal variation. The $b$-value continuously decreases when approaching the earthquake and then reaches the minimum $b$-value, which is 9 hours earlier than the occurrence time of the earthquake. Moreover, the continuous downward trend of the $b$-value can be used as an early warning of earthquakes.

[Figure]

Fig.14: AE parameters and *b*-value when approaching the major earthquake occurrence. t$_{crit}$ represents the critical time as predicted by the natural-time analysis, t_$_{EQ.1}$ depicts the occurrence time of earthquake EQ.1. >>

*(6) The presentation of the results should be more concise but also more informative. For example, Fig. 9 does not add any additional information and should be omitted, while the cumulative distribution would be better to be included in Fig. 8 as an additional curve. Additionally, Figures 11, 12 and 13 actually present the same information and should be incorporated in a single figure.*

The figures of the manuscript have been revised as follows:

<<…

[Figure]

Figure 8: AE rate and cumulated distribution versus seismic sequence.

[Figure]

Figure 10: Multi-modal Gaussian distribution of earthquakes (a), AE (b), superposition (c) and the predicted results (d),(e)

…>>

*(7) Some "technical" issues about b-value and natural time analysis should be included, for example, the time window and overlapping that was used, etc. Furthermore, natural time analysis*

*should be carried out also for the other main event, and generally, for all major events when a broader period will be considered.*

By considering the Reviewer's suggestion, the text of the manuscript has been revised as follows:

<<…The *b*-value is the negative slope of the GR law straight line, which is fitted by the least squares method. In this study, the temporal variation of the *b*-value is estimated by the moving event window method. A number of events, *N*, equal to 400, and a time window step of 200 events are adopted for the evaluation of the *b*-value temporal variation. >>

<< Recently, natural time analysis has been applied to identify the imminent failure of materials and structures (Loukidis et al., 2020; Ferreira et al., 2022a and 2022b; Triantis et al., 2023). Natural-time series transform time series into the natural-time domain, neglecting the time intervals of consecutive events and only considering the order and energy of occurrence. Based on the time-series analysis of *N* events read in a new time domain, namely the natural time, $\chi$ , a method to identify critical states was developed (Varotsos et al., 2011 and 2013). The variance $\kappa_1$ of the natural time is defined as:

$$\kappa_1 = \sum_{k=1}^{N} p_k \chi_k{}^2 - \left( \sum_{k=1}^{N} p_k \chi_k \right)^2 = \langle \chi^2 \rangle - \langle \chi \rangle^2 , \tag{4}$$

where $\chi_K = K / N$ is the normalized index of of energy $Q_k$ (related AE energy), and $P_K = Q_K / \sum_{i=1}^{N} Q_i$ is a probability distribution of the discrete variable $\chi_K$ . When $\kappa_1$ converges to 0.07, the critical state is imminent…>>